# Trends in proton pump inhibitor use, reflux esophagitis, and various upper gastrointestinal symptoms from 2010 to 2019 in Japan

**Nobutake Yamamichi**[1,2]*, **Takeshi Shimamoto**[1,3], **Yu Takahashi**[2], **Mami Takahashi**[1], **Chihiro Takeuchi**[2], **Ryoichi Wada**[3], **Mitsuhiro Fujishiro**[2]

1 Center for Epidemiology and Preventive Medicine, The University of Tokyo Hospital, Bunkyo-ku, Tokyo, Japan, 2 Department of Gastroenterology, Graduate School of Medicine, The University of Tokyo, Bunkyo-ku, Tokyo, Japan, 3 Kameda Medical Center Makuhari CD-2, Mihama-ku, Chiba-City, Japan

* NobutakeYamamichi@gmail.com

## Abstract

The increasing usage of proton pump inhibitors (PPIs) has been reported worldwide, but information on PPI use in East Asia is inadequate. This study aimed to examine the trends in PPI use in Japan, along with the changes in histamine $H_2$ receptor antagonist ($H_2$RA) use, disease rate of reflux esophagitis, and the prevalence of upper gastrointestinal symptoms. We analyzed 217,712 healthy subjects (127,607 men and 90,105 women; 51.4 ± 9.7 years old) participating in the health check program from 2010 to 2019. Various upper gastrointestinal symptoms were evaluated using the frequency scale for the symptoms of gastroesophageal reflux disease (FSSG) questionnaire. Reflux esophagitis was diagnosed by esophageal erosion using the Los Angeles classification grades A, B, C, and D. From 2010 to 2019, the percentage of PPI users increased markedly from approximately 1.8% to 5.3%, whereas that of $H_2$RA users decreased gradually from approximately 2.5% to 1.9%. The use of all classical types of PPIs (omeprazole, lansoprazole, rabeprazole, and esomerazole) and a new type of PPI, a potassium-competitive acid blocker (vonoprazan), greatly increased during the 10 years. An upward trend in the prevalence of reflux esophagitis was observed from 2010 to 2015, but not from 2016 to 2019, indicating that the monotonic rising prevalence of reflux disease stopped in the middle of the 2010s in Japan. In contrast, various upper gastrointestinal symptoms significantly improved between 2010 and 2019. All 12 FSSG symptoms of PPI users were significantly worse than those of non-PPI users, suggesting that PPIs still cannot completely control upper gastrointestinal symptoms. In conclusion, this study revealed a significant increase in PPI use and a slight decrease in $H_2$RA use from 2010 to 2019. Despite a plateau in the prevalence of reflux esophagitis and considerable improvement in various upper gastrointestinal symptoms, PPI use has continued to increase in Japan.

**Data Availability Statement:** All the raw data were uploaded as Supporting information file.

**Funding:** The author(s) received no specific funding for this work.

**Competing interests:** The authors have declared that no competing interests exist.

## Introduction

Proton pump inhibitors (PPIs) are the most common acid-suppressive medications that inhibit the final step of acid suppression [1]. PPIs are usually prescribed for the treatment of various disorders such as gastroesophageal reflux disease (GERD) [2, 3], peptic ulcer diseases [4], *Helicobacter pylori* (*H. pylori*) eradication therapy [5], dyspepsia [6, 7], Barrett's esophagus [8], and eosinophilic esophagitis [9]. Similar to many developed countries, GERD is the most common acid-related disease in Japan [10]. In fact, our recent study showed that the prevalence of reflux esophagitis is much higher than that of peptic ulcers, which is partly due to the decreased prevalence of infection with *H. pylori* [11]. Consequently, PPIs are the most frequently used treatment for GERD in Japan.

As many previous studies have reported [12–14], the prevalence of severe reflux esophagitis is low in the Japanese population. Most cases of reflux esophagitis are diagnosed as LA-A or LA-B grades according to the Los Angeles classification [15]. Concerning the therapy for reflux esophagitis, LA-C and LA-D grades are considered as conclusive of GERD [16] and should therefore be treated with PPI. In contrast, the therapeutic plan for reflux esophagitis of grades LA-A and LA-B is determined by considering the accompanying upper gastrointestinal symptoms. Therefore, in Japan, the major reason for continuous PPI use is not to heal esophageal erosion but to control upper gastrointestinal symptoms. It is also known that the disease rate of non-erosive reflux disease (NERD) is higher than that of reflux esophagitis in Japan [10, 17]. Therefore, controlling upper gastrointestinal symptoms (especially acid reflux) is the major reason for PPI use in Japan.

Although several studies have indicated the risk of long-term PPI use [18, 19], increasing PPI usage has been reported in various regions of the world [20–23]. However, information regarding PPI use in East Asia is currently inadequate. Therefore, in this study, we examined the trend of PPI use from 2010 to 2019 using large-scale cohort data from generally healthy individuals in Japan. Along with the PPI use, we also examined the use of histamine $H_2$ receptor antagonists ($H_2$RA), the trend in reflux esophagitis, and the change in upper gastrointestinal symptoms during the 10 years. We would like to understand the current use of gastric acid suppressants together with the changes in upper gastrointestinal disorders based on large-scale cohort data of the general population in Japan.

## Materials and methods

### Study subjects

A total of 223,793 healthy subjects who underwent medical check-ups at Kameda Medical Center Makuhari (Chiba, Japan) were prospectively enrolled from 2010 to 2019. Subjects with insufficient data for analysis were excluded from the study. This study was conducted in accordance with the World Medical Association Declaration of Helsinki and approved by the ethics committee of the Kameda Medical Center (approval no. 17–705). Written informed consent was obtained from all study participants and all data were fully anonymized before access by the researchers.

### Frequency scale for the symptoms of GERD (FSSG) questionnaire

The FSSG is a widely used questionnaire for diagnosing GERD-related upper gastrointestinal symptoms, in which 12 questions cover various acid reflux and gastric dysmotility symptoms [24, 25]. The total FSSG score ranges from 0 to 48, and a score > 7 indicates the presence of abnormal GERD-related upper gastrointestinal symptoms [26]. In this study, the score ratio of each upper gastrointestinal symptom for PPI users was calculated by dividing the average PPI

users' scores by the average non-PPI users' scores. These 12 upper gastrointestinal symptoms were further classified into five categories of symptoms: (A) heartburn, (B) dyspepsia, (C) acid regurgitation, (D) pharyngo-upper esophageal discomfort, and (E) fullness while eating, which was based on a previous hierarchical cluster analysis (Ward's method with Euclidean distances) [27].

### Diagnosis of reflux esophagitis

Reflux esophagitis (erosive GERD) was diagnosed according to the Los Angeles classification as in our previous reports [11, 17]. Grades A, B, C, and D (LA-A, B, C, and D) of esophageal erosion were diagnosed as the presence of reflux esophagitis. In this study, grade M (LA-M; minimal changes to the mucosa) was not considered to indicate the presence of reflux esophagitis [28].

### Statistical analyses

We used JMP 10 software or SAS 9.1.3 (SAS Institute Inc. Cray, NC, USA) for statistical analyses. To compare the average score of 12 FSSG symptoms between PPI users and non-PPI users, Student's t-test was used and p values <0.05 were considered as statistically significant.

## Results

### Trend in PPI and $H_2RA$ uses from 2010 to 2019 in Japan

After excluding subjects with insufficient data for analysis, a total of 217,712 subjects (127,607 men and 90,105 women; 51.4 ± 9.7 years old; range, 20–93 years) from 2010 to 2019 were analyzed. As shown in Fig 1(a), the rate of those who used gastric acid suppressants (PPI or $H_2RA$) increased during the 10 years. The percentage of PPI users markedly increased from approximately 1.8% in 2010 to 5.3% in 2019. On the other hand, the percentage of $H_2RA$ users gradually decreased from approximately 2.5% (2010) to 1.9% (2019).

Fig 1(b) shows the breakdown of PPI use in the last decade. In Japan, three classical PPIs (omeprazole, lansoprazole, and rabeprazole) were available in 2010. Esomeprazole, another classical PPI, was first used in 2011 [29]. Vonoprazan, a new type of PPI called potassium-competitive acid blocker, was first used in 2015 [30]. Notably, the use of esomeprazole and vonoprazan has monotonically increased since their release (Fig 1(b)).

### Trend in reflux esophagitis from 2010 to 2019 in Japan

Subsequently, we evaluated the prevalence of reflux esophagitis from 2010 to 2019 since PPIs are most frequently used for the treatment of GERD. By summarizing many previous studies from Japan, Fujiwara *et al.* reported an apparent increase in the disease rate of GERD [10]. Our recent reports also showed that the prevalence of reflux esophagitis has markedly increased during 25 years (from 1990 to 2015), which was partly due to the reduced rate of infection with *H. pylori* [11].

In this study, we examined the recent trend in reflux esophagitis in our cohort from 2010 to 2019. As shown in Fig 2, an upward trend in the prevalence of reflux esophagitis was observed from 2010 to 2015. However, this increase was not observed from 2016 to 2019. Our data indicated that the monotonic increase in reflux esophagitis in Japan finally stopped in the middle of the 2010s.

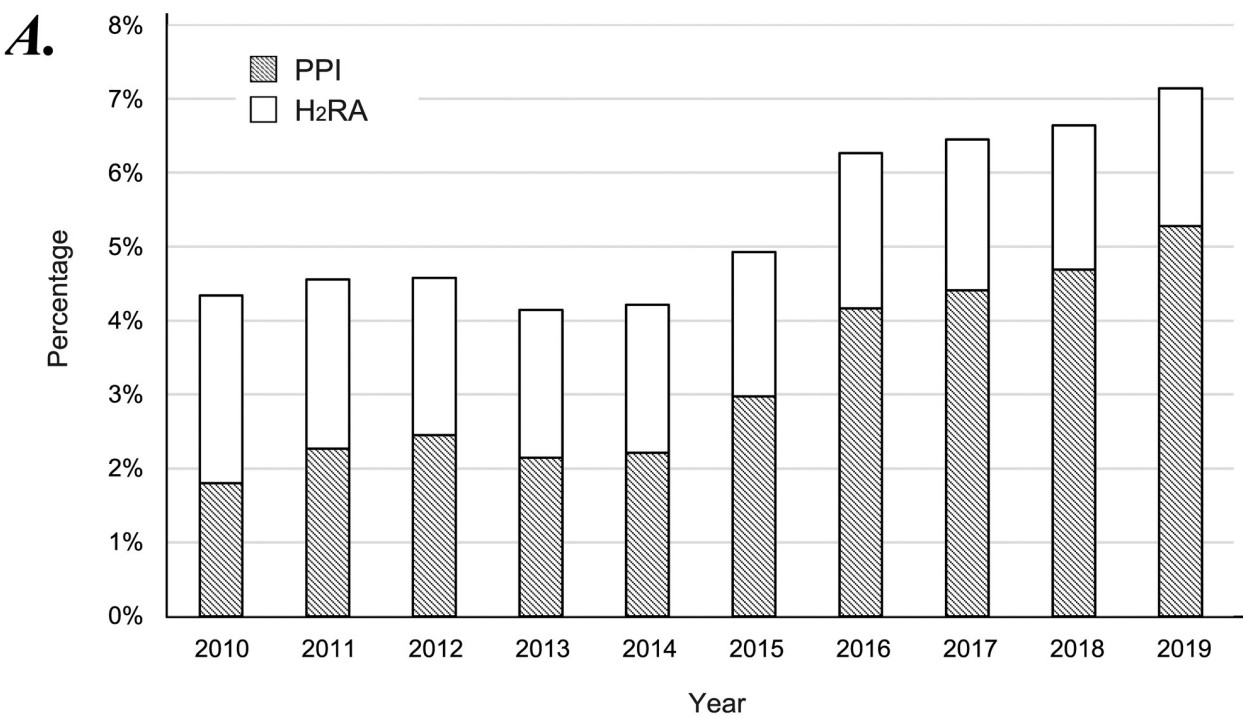

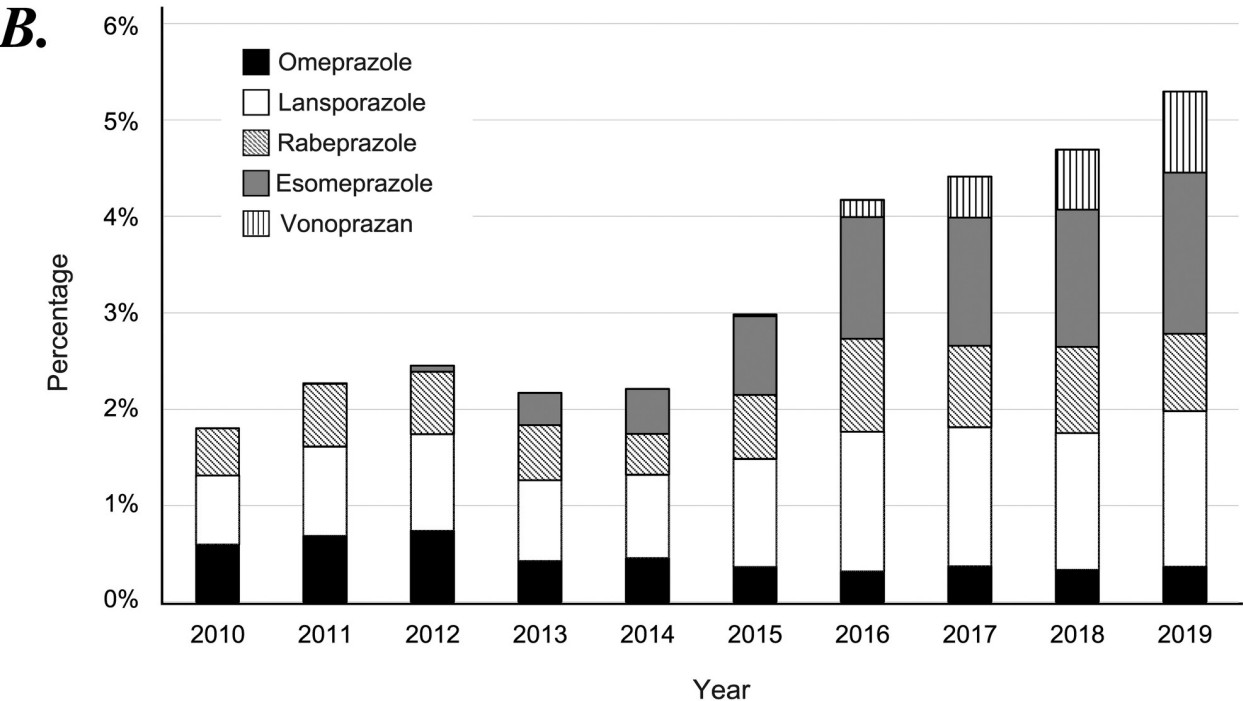

**Fig 1. Trend in PPI and H$_2$RA uses from 2010 to 2019 in Japan.** (a) Changes in the percentages of PPI and H$_2$RA users from 2010 to 2019 based on the data of generally healthy individuals in Japan. (b) Changes in the rate of the use of five types of PPI from 2010 to 2019 in Japan. PPI, proton pump inhibitor; H$_2$RA, histamine H$_2$ receptor antagonist.

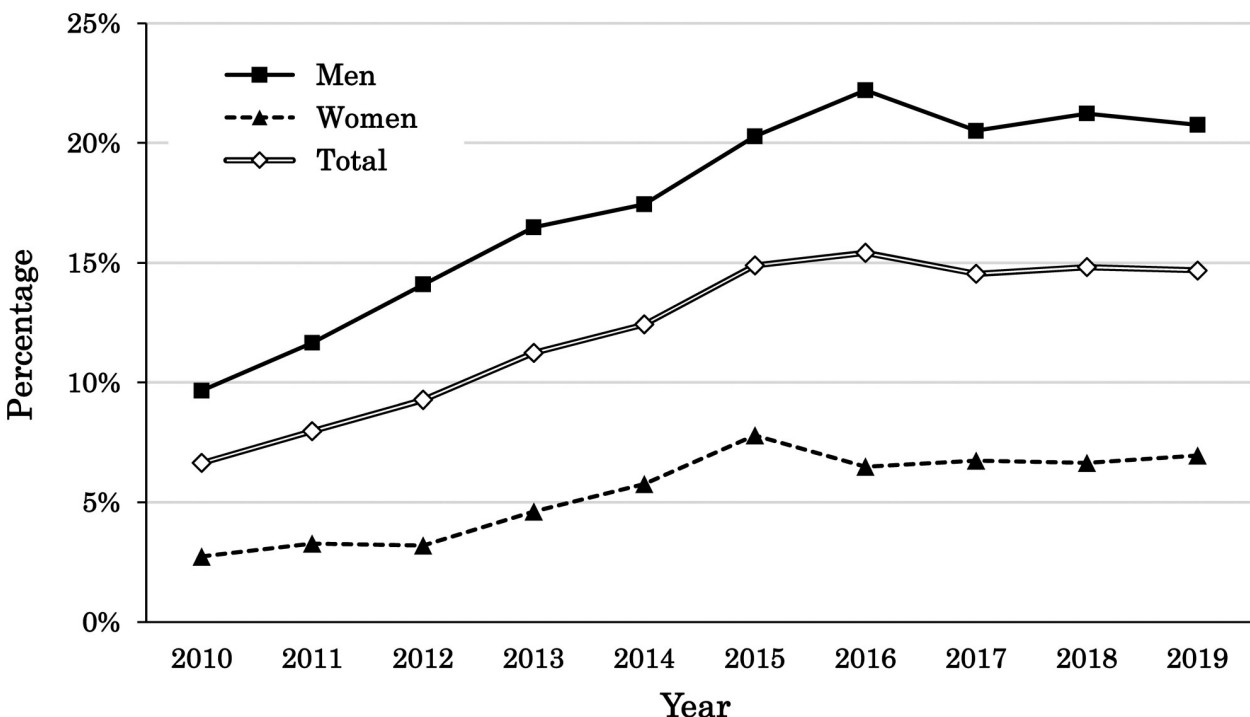

**Fig 2. Changes in the prevalence of reflux esophagitis (men, women, and total) from 2010 to 2019 in Japan.**

### Trend in upper gastrointestinal symptoms from 2010 to 2019 in Japan

Next, we examined the changes in the upper gastrointestinal symptoms from 2010 to 2019 using the FSSG questionnaire [26]. As shown in Fig 3, the total FSSG score gradually decreased over the last decade. Upon closer inspection, all the scores of the 12 questions decreased during the 10 years. According to the diagnosis by the FSSG score (a score of more than seven was considered abnormal), the percentage of subjects with abnormal upper gastrointestinal symptoms notably declined from approximately 23.5% (2010) to 15.5% (2019).

As described in the Introduction, the major reason for the continued use of PPIs is to control upper gastrointestinal symptoms [12–14, 16]. Therefore, we examined the association between various upper gastrointestinal symptoms and PPI use over the period of 10 years. As shown in Fig 4, all 12 upper gastrointestinal symptoms in PPI users were significantly worse than those in non-PPI users in our cohort ($p < 0.05$ for all 12 symptoms from 2010 to 2019). Furthermore, Fig 4 also shows that all 12 upper gastrointestinal symptoms improved from 2010 to 2019 for both PPI and non-PPI users. These results imply that PPI users still suffer from diverse upper gastrointestinal symptoms despite PPI use. Even today, there are various upper gastrointestinal symptoms that are not well controlled by PPIs.

### Association between the use of PPI and the 12 upper gastrointestinal symptoms

Finally, we investigated the association between PPI use and the 12 upper gastrointestinal symptoms (Fig 5). As shown in Fig 5(a), the average scores of the 12 upper gastrointestinal symptoms for PPI users and non-PPI users varied considerably. The symptom scores of PPI users were significantly greater than those of non-PPI users ($p < 0.05$ for all the 12 symptoms),

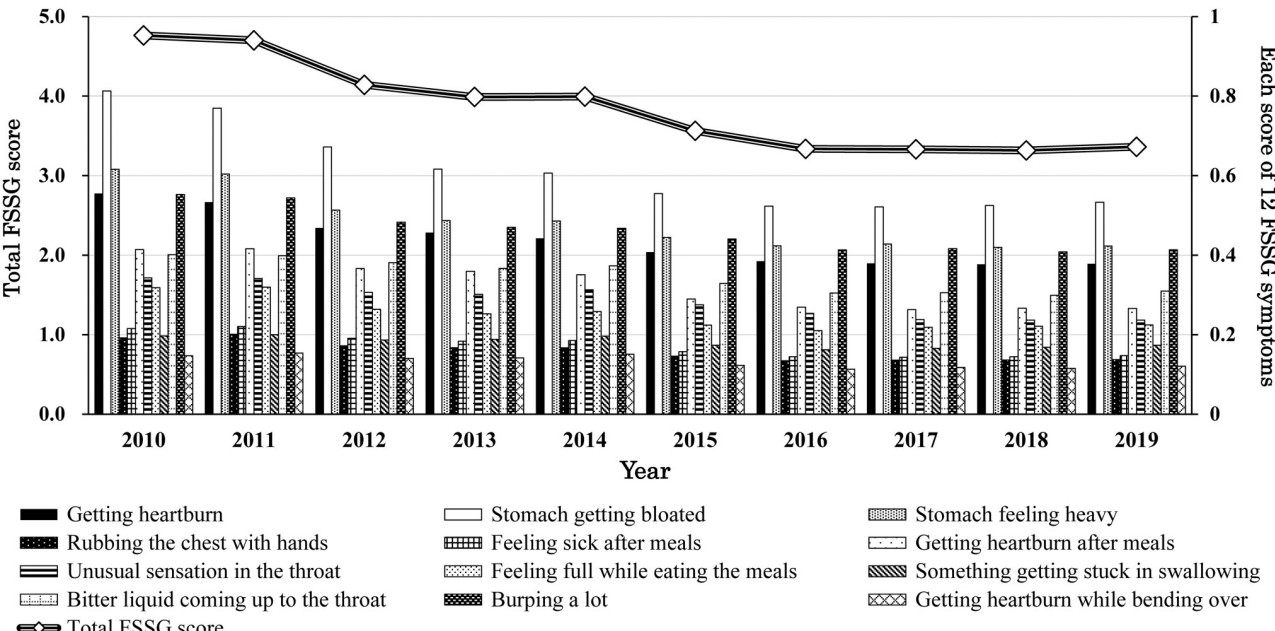

**Fig 3. Changes in the total FSSG score and 12 FSSG symptoms from 2010 to 2019 in Japan.** The left axis (from 0 to 5) is for the total FSSG score, and the change of total FSSG score is shown by line graph. The right axis (from 0 to 1) is for each of the 12 FSSG scores, which are shown by bar graphs from 2010 to 2019. FSSG, frequency scale for the symptoms of gastroesophageal reflux disease.

suggesting that PPI is used to control various upper gastrointestinal symptoms but not all symptoms entirely.

The score ratios of the 12 upper gastrointestinal symptoms (PPI users vs. non-PPI users) were diverse and ranged widely from 1.23–2.92. Based on the five categories of symptoms derived from the hierarchical cluster analysis of FSSG symptoms [27], all symptom score ratios are illustrated in Fig 5(b). At a glance, most symptom score ratios for "heartburn" and "acid regurgitation" categories were high except for "burping a lot"; a score ratio of "burping a lot" was significantly lower than those of other four symptoms (p < 0.05). Similarly, most symptom score ratios for "dyspepsia" and "fullness while eating" categories were low except for "rubbing the chest with hands"; a score ratio of "rubbing the chest with hands" was significantly higher than those of other four symptoms (p < 0.05). As the hierarchical cluster analysis of the five symptom categories can reflect a similar etiology or mechanism of various upper gastrointestinal symptoms belonging to the same category, the significant difference in PPI use among the five symptom categories should suggest the different involvement of gastric acid upon them.

To grasp various upper gastrointestinal symptoms more simply, we further classified the 12 symptoms into three categories: "heartburn or acid regurgitation (GERD-related gastrointestinal symptoms)", "dyspepsia or fullness while eating (GERD-unrelated symptoms)", and "pharyngo-upper esophageal discomfort (symptoms of pharyngo-upper esophageal area)". As shown in Fig 5(c), the ratios of FSSG score between PPI users and non-PPI users were 2.13 for "heartburn or acid regurgitation", 1.47 for "dyspepsia or fullness while eating (GERD-unrelated symptoms)", and 1.73 for "pharyngo-upper esophageal discomfort". Our data showed that PPIs are most frequently used for GERD-related symptoms, but also indicated that they are sometimes used for other upper gastrointestinal symptoms with little relation to GERD.

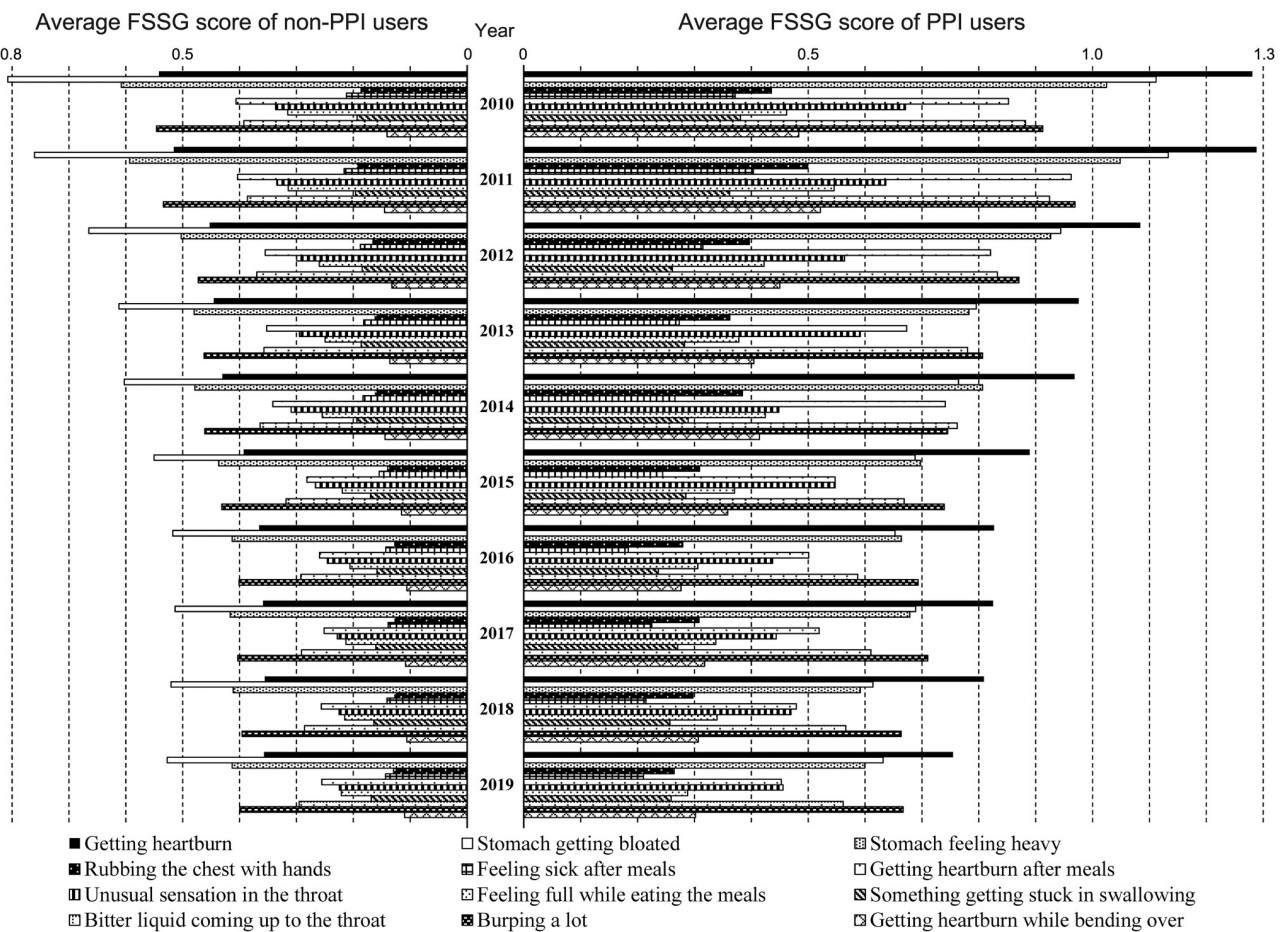

**Fig 4. Changes in the 12 gastrointestinal symptoms based on the FSSG scores from 2010 to 2019 among generally healthy individuals in Japan.** The left half shows the average score of 12 FSSG symptoms in non-PPI users, and the right half shows the average score of 12 FSSG symptoms in PPI users. FSSG, frequency scale for the symptoms of gastroesophageal reflux disease.

## Discussion and conclusion

Using large-scale cohort data of generally healthy individuals in Japan, our study clearly showed a marked increase in PPI use in Japan, which is accompanied by a gradual decrease in $H_2RA$ use. The increasing and frequent use of PPIs has been reported in many countries such as the United States [31], France [32], Hungary [23], and Iceland [21]. In comparison with Western countries, the percentage of PPI users is still lower in Japan, but the monotonous increase and PPI use of more than 5% in a generally healthy population are noteworthy trends in Japan (Fig 1).

Recently, Sakata et al. reported an increasing use of PPIs based on prescription data from 2009 to 2016 [33]. Although there were some differences in the breakdown of the agents used, Sakata's report and our study both showed an upward trend in the PPI use in Japan. By analyzing the data of patients who were prescribed PPIs in the hospital, Sakata showed an increasing number of patients taking PPIs together with antithrombotic drugs [33]. In contrast, our study analyzed the data of healthy individuals participating in a health check program. Therefore, compared with Sakata's study, the percentage of individuals who used antithrombotic drugs was much lower in our study population, and most PPI users in this study had bothersome upper gastrointestinal symptoms and/or reflux esophagitis. Despite a plateau in the increased

**A.**

| FSSG Score | Getting heartburn | | | Stomach getting bloated | | | Stomach feeling heavy | | | Rubbing the chest with hands | | |
|---|---|---|---|---|---|---|---|---|---|---|---|---|
| | Total | Use of PPI | Non use of PPI | Total | Use of PPI | Non use of PPI | Total | Use of PPI | Non use of PPI | Total | Use of PPI | Non use of PPI |
| 0 | 70.7% | 49.7% | 71.4% | 63.2% | 59.4% | 63.3% | 67.7% | 58.1% | 68.1% | 89.1% | 79.6% | 89.4% |
| 1 | 17.7% | 20.5% | 17.6% | 18.8% | 16.3% | 18.8% | 19.5% | 18.9% | 19.5% | 7.1% | 10.9% | 7.0% |
| 2 | 9.5% | 21.4% | 9.1% | 13.4% | 16.9% | 13.2% | 10.2% | 16.9% | 10.0% | 3.1% | 7.0% | 2.9% |
| 3 | 1.9% | 7.0% | 1.7% | 3.9% | 5.7% | 3.9% | 2.2% | 5.0% | 2.1% | 0.7% | 2.3% | 0.7% |
| 4 | 0.2% | 1.5% | 0.2% | 0.8% | 1.7% | 0.8% | 0.3% | 1.2% | 0.3% | 0.1% | 0.3% | 0.1% |
| Average | 0.43 | 0.9 | 0.42 | 0.61 | 0.74 | 0.6 | 0.48 | 0.72 | 0.47 | 0.16 | 0.33 | 0.15 |

| FSSG Score | Feeling sick after meals | | | Getting heartburn after meals | | | Unusual sensation in the throat | | | Feeling full while eating the meals | | |
|---|---|---|---|---|---|---|---|---|---|---|---|---|
| | Total | Use of PPI | Non use of PPI | Total | Use of PPI | Non use of PPI | Total | Use of PPI | Non use of PPI | Total | Use of PPI | Non use of PPI |
| 0 | 86.7% | 82.7% | 86.8% | 76.3% | 63.3% | 76.8% | 83.5% | 73.9% | 83.8% | 82.6% | 77.8% | 82.8% |
| 1 | 10.1% | 11.4% | 10.1% | 16.6% | 19.3% | 16.5% | 8.6% | 11.1% | 8.5% | 11.3% | 11.8% | 11.3% |
| 2 | 2.6% | 4.7% | 2.6% | 5.9% | 13.3% | 5.6% | 5.1% | 8.4% | 5.0% | 4.8% | 7.5% | 4.7% |
| 3 | 0.5% | 1.1% | 0.5% | 1.1% | 3.6% | 1.0% | 2.0% | 4.7% | 1.9% | 1.0% | 2.1% | 1.0% |
| 4 | 0.1% | 0.2% | 0.1% | 0.1% | 0.6% | 0.1% | 0.8% | 2.0% | 0.8% | 0.2% | 0.7% | 0.2% |
| Average | 0.17 | 0.25 | 0.17 | 0.32 | 0.59 | 0.31 | 0.28 | 0.5 | 0.27 | 0.25 | 0.36 | 0.24 |

| FSSG Score | Something getting stuck in swallowing | | | Bitter liquid coming up to the throat | | | Burping a lot | | | Getting heartburn while bending over | | |
|---|---|---|---|---|---|---|---|---|---|---|---|---|
| | Total | Use of PPI | Non use of PPI | Total | Use of PPI | Non use of PPI | Total | Use of PPI | Non use of PPI | Total | Use of PPI | Non use of PPI |
| 0 | 87.1% | 82.0% | 87.2% | 74.3% | 57.9% | 74.9% | 72.5% | 60.6% | 72.9% | 90.7% | 78.4% | 91.1% |
| 1 | 8.9% | 11.1% | 8.8% | 18.5% | 23.2% | 18.3% | 14.8% | 17.2% | 14.8% | 6.3% | 11.5% | 6.1% |
| 2 | 3.2% | 4.9% | 3.1% | 5.8% | 14.1% | 5.6% | 8.3% | 12.7% | 8.1% | 2.2% | 7.3% | 2.1% |
| 3 | 0.8% | 1.6% | 0.7% | 1.2% | 3.9% | 1.1% | 3.2% | 6.4% | 3.1% | 0.6% | 2.3% | 0.6% |
| 4 | 0.1% | 0.4% | 0.1% | 0.1% | 0.8% | 0.1% | 1.2% | 3.1% | 1.1% | 0.1% | 0.6% | 0.1% |
| Average | 0.18 | 0.28 | 0.18 | 0.34 | 0.67 | 0.33 | 0.46 | 0.74 | 0.45 | 0.13 | 0.35 | 0.12 |

**B.**

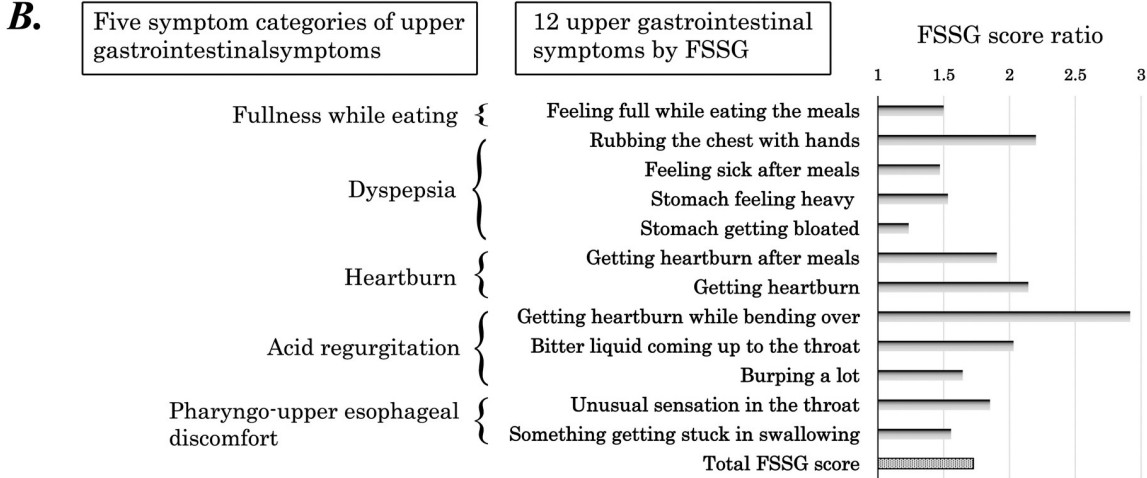

**C.**

| Average FSSG Score | Dyspepsia / Fullness while eating | | | Heartburn / Acid regurgitation | | | Pharyngo-upper esophageal discomfort | | |
|---|---|---|---|---|---|---|---|---|---|
| | Total | Use of PPI | Non use of PPI | Total | Use of PPI | Non use of PPI | Total | Use of PPI | Non use of PPI |
| 0 | 77.86 | 71.54 | 78.07 | 78.01 | 62.30 | 78.55 | 85.27 | 77.91 | 85.53 |
| 1 | 13.36 | 13.84 | 13.35 | 14.76 | 18.61 | 14.63 | 8.74 | 11.10 | 8.66 |
| 2 | 6.81 | 10.59 | 6.68 | 5.86 | 14.02 | 5.58 | 4.14 | 6.64 | 4.05 |
| 3 | 1.67 | 3.23 | 1.61 | 1.21 | 4.21 | 1.11 | 1.37 | 3.15 | 1.31 |
| 4 | 0.31 | 0.80 | 0.29 | 0.16 | 0.87 | 0.13 | 0.48 | 1.22 | 0.46 |
| Average | 0.33 | 0.48 | 0.33 | 0.31 | 0.63 | 0.30 | 0.23 | 0.39 | 0.23 |
| FSSG symptom score ratio (PPI users vs non-PPI users) | 1.47 | | | 2.13 | | | 1.73 | | |

**Fig 5. The relationship between the use of PPI and the 12 upper gastrointestinal symptoms by FSSG.** (a) Detailed data of the 12 upper gastrointestinal symptoms of PPI users and non-PPI users. (b) The score ratio of the 12 upper gastrointestinal symptoms of PPI users in comparison with non-PPI users. (c) Detailed data and the score ratio of three upper gastrointestinal symptom categories. PPI, proton pump inhibitor; FSSG, frequency scale for the symptoms of gastroesophageal reflux disease.

prevalence of reflux esophagitis (Fig 2) and considerable improvement in various upper gastrointestinal symptoms (Fig 3), it should be worthy of mention that PPI use has still been increasing in Japan.

The prevalence of infections with *H. pylori* has been rapidly decreasing not only in Japan but also worldwide [11, 34], which can lead to a reduced prevalence in mucosal atrophy, increase in intragastric pH, and rising disease rate of reflux disease [35, 36]. We believe that the declining rate of infection with *H. pylori* is one of the reasons for the increased PPI use. However, as shown in Fig 2, the increasing trend in reflux esophagitis finally stopped in the middle of the 2010s. In addition, various upper gastrointestinal symptoms in the Japanese population have improved over the last decade (Figs 3 and 4). Our data cannot adequately explain why PPI use continues to increase.

The discrepancy between the ratio of the 12 upper gastrointestinal symptoms scores (PPI users vs. non-PPI users; Fig 4) and the hierarchical cluster analysis of the five categories of symptoms (Fig 5) indicated that PPI is prescribed for various upper gastrointestinal symptoms and not always used for typical acid-related situations. As expected, our results (Fig 5(b) and 5 (c)) showed that PPI is most frequently used and obviously effective for GERD-related symptoms (heartburn or acid regurgitation). However, our results (Figs 4 and 5) also indicated that PPI is sometimes used for GERD-unrelated symptoms (dyspepsia or fullness while eating), as all and total symptom score ratios (PPI users vs. non-PPI users) were more than 1 with statistical significance (Fig 5(b) and 5(c)). Although the mechanism of PPIs upon dyspepsia symptoms has not been elucidated, a recent large-scale meta-analysis showed that PPIs are effective for the treatment of functional dyspepsia [37]. Our results derived from the data of healthy people in Japan suggest that PPI can control GERD-unrelated upper gastrointestinal symptoms to some extent, which is consistent with the result of the meta-analysis evaluating 25 randomized controlled trials worldwide [37].

As described in the Introduction, many studies have mentioned the risk of long-term PPI use [18, 19]. In particular, safety data on the long-term use of vonoprazan are insufficient because this new and highly effective gastric acid suppressant was prescribed only seven years ago. Following careful consideration of the effects and risks of PPIs, we should continue to assess the necessity of PPI use based on multiple epidemiological factors such as the disease rate of reflux esophagitis, prevalence of chronic infection with *H. pylori*, trend in various upper gastrointestinal symptoms, and the increasing use of antithrombotic drugs.

In conclusion, large-scale data of 217,712 generally healthy individuals in Japan revealed a significant increase in PPI use accompanied by a slight decrease in $H_2RA$ use from 2010 to 2019. Despite a plateaued increase in the prevalence of reflux esophagitis in the middle of the 2010s and considerable improvement in various upper gastrointestinal symptoms from 2010 to 2019, PPI use has continued to increase in Japan. All upper gastrointestinal symptoms in PPI users were significantly worse than those in non-PPI users, indicating that PPIs still cannot presently control all upper gastrointestinal symptoms.

## Supporting information

**S1 Data.**
(XLSX)

## Acknowledgments

We are grateful to Mr. Minoru Okada (Kameda Medical Center Makuhari, Chiba-shi, Chiba, Japan) for assistance with establishment and maintenance of the study database.

## Author Contributions

**Conceptualization:** Nobutake Yamamichi, Takeshi Shimamoto.

**Data curation:** Nobutake Yamamichi, Takeshi Shimamoto, Yu Takahashi, Mami Takahashi.

**Investigation:** Nobutake Yamamichi, Takeshi Shimamoto, Mami Takahashi.

**Supervision:** Ryoichi Wada, Mitsuhiro Fujishiro.

**Validation:** Takeshi Shimamoto.

**Writing – original draft:** Nobutake Yamamichi.

**Writing – review & editing:** Yu Takahashi, Chihiro Takeuchi.

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
