## [Decision Letter · Decision Letter 0]

11 May 2022

PONE-D-22-02306Trends in proton pump inhibitor use, reflux esophagitis, and various upper gastrointestinal symptoms from 2010 to 2019 in JapanPLOS ONE

Dear Dr. Yamamichi,

Thank you for submitting your manuscript to PLOS ONE. After careful consideration, we feel that it has merit but does not fully meet PLOS ONE’s publication criteria as it currently stands. Therefore, we invite you to submit a revised version of the manuscript that addresses the points raised during the review process.

We look forward to receiving your revised manuscript.

Kind regards,

Rasa Zarnegar, MD

Academic Editor

PLOS ONE

Journal Requirements:

Additional Editor Comments:

I think this manuscript has value.

Please respond to the reviewer comments below.

Reviewers' comments:

Reviewer's Responses to Questions

**Comments to the Author**

1. Is the manuscript technically sound, and do the data support the conclusions?

Reviewer #1: Partly

2. Has the statistical analysis been performed appropriately and rigorously? 

Reviewer #1: Yes

3. Have the authors made all data underlying the findings in their manuscript fully available?

Reviewer #1: Yes

4. Is the manuscript presented in an intelligible fashion and written in standard English?

Reviewer #1: Yes

5. Review Comments to the Author

Reviewer #1: I applaud the Authors for a thorough review and the impressive patient population included in the study. The manuscript presents a decade of experience in prescribing antacid medications in Japan which mirrors the international trend which has seen an increase in the prescription of PPIs worldwide. Importantly it sheds a light on the issue of overprescription of PPIs for the wrong symptoms (such as the most common: bloating). However before publication several points need to be addressed:

1) The tables appear to be overly convoluted and difficult to read, mostly because of an abundance of duplicate entries from the FSSG questionnaire (Heartburn, heartburn while bending, heartburn after meals). I wonder if a simplification of the symptoms categories would be possible including clustering similar symptoms together (heartburn) and/or removing symptoms which are not very specific to GERD (feeling sick after meals) as tables are very difficult to read in their present form. Perhaps splitting symptoms into typical (HB, regurgitation, chest pain and dysphagia) and atypical could help. All the 12 entries could be left as is in table 5.

2) It would be interesting to know the schedule and indications for EGD surveillance and whether the authors subscribe to ASGE guidelines

3) It seems like the most common symptom the patient report is bloating, which is not usually improved by PPIs administration. There is however an improvement in patients reporting this symptom amongst PPI takers. This should be expanded upon in the discussion.

4) It would be interesting to know how many of the patients underwent anti-reflux surgery for their symptoms. I know its not very popular in Japan but perhaps some of the patients underwent this route.

5) each figure should have a description for easier interpretability

6) given the detail of the database it would be good to know if authors also collected other medical history hinting to PPIs side effects such as PNA, C. Difficlile Colitis, fractures and esophageal cancer rates.

6. PLOS authors have the option to publish the peer review history of their article (what does this mean?). If published, this will include your full peer review and any attached files.

Reviewer #1: **Yes: **Filippo Filicori, MD

---

## [Author Response · Author response to Decision Letter 0]

26 May 2022

All the responses are described in the uploaded MS-word file (response to reviewers). 

Please see the file whose name is "Rebuttal 01_PLOSONE.doc".

---

## [Editor Report · Decision Letter 1]

8 Jun 2022

Trends in proton pump inhibitor use, reflux esophagitis, and various upper gastrointestinal symptoms from 2010 to 2019 in Japan

PONE-D-22-02306R1

Dear Dr. Yamamichi,

We’re pleased to inform you that your manuscript has been judged scientifically suitable for publication and will be formally accepted for publication once it meets all outstanding technical requirements.

Kind regards,

Rasa Zarnegar, MD

Academic Editor

PLOS ONE
---

## [Editor Report · Acceptance letter]

10 Jun 2022

PONE-D-22-02306R1 

Trends in proton pump inhibitor use, reflux esophagitis, and various upper gastrointestinal symptoms from 2010 to 2019 in Japan 

Dear Dr. Yamamichi:

I'm pleased to inform you that your manuscript has been deemed suitable for publication in PLOS ONE. Congratulations! Your manuscript is now with our production department. 

Kind regards, 

on behalf of

Dr. Rasa Zarnegar 

Academic Editor

PLOS ONE